# Surface-Active Compounds Produced by Microorganisms: Promising Molecules for the Development of Antimicrobial, Anti-Inflammatory, and Healing Agents

**DOI:** 10.3390/antibiotics11081106

**Published:** 2022-08-16

**Authors:** Jéssica Araujo, Joveliane Monteiro, Douglas Silva, Amanda Alencar, Kariny Silva, Lara Coelho, Wallace Pacheco, Darlan Silva, Maria Silva, Luís Silva, Andrea Monteiro

**Affiliations:** 1Rede de Biodiversidade e Biotecnologia da Amazônia Legal, BIONORTE, São Luís 65055-310, MA, Brazil; 2Laboratório de Microbiologia Aplicada, Universidade CEUMA, São Luís 65075-120, MA, Brazil; 3Laboratório de Ciências do Ambiente, Universidade CEUMA, São Luís 65075-120, MA, Brazil; 4Laboratório de Patogenicidade Microbiana, Universidade CEUMA, São Luís 65075-120, MA, Brazil

**Keywords:** surface-active compounds, biosurfactants, bioemulsifier, biological properties, biotechnology

## Abstract

Surface-active compounds (SACs), biomolecules produced by bacteria, yeasts, and filamentous fungi, have interesting properties, such as the ability to interact with surfaces as well as hydrophobic or hydrophilic interfaces. Because of their advantages over other compounds, such as biodegradability, low toxicity, antimicrobial, and healing properties, SACs are attractive targets for research in various applications in medicine. As a result, a growing number of properties related to SAC production have been the subject of scientific research during the past decade, searching for potential future applications in biomedical, pharmaceutical, and therapeutic fields. This review aims to provide a comprehensive understanding of the potential of biosurfactants and emulsifiers as antimicrobials, modulators of virulence factors, anticancer agents, and wound healing agents in the field of biotechnology and biomedicine, to meet the increasing demand for safer medical and pharmacological therapies.

## 1. Introduction

Microorganisms can produce several surface-active compounds (SACs) with hydrophilic and hydrophobic moieties. These structural features allow them to interact with the surface and interfacial tensions, form micelles, and emulsify immiscible substances [1,2].

Biosurfactants (BSs) and bioemulsifiers (BEs) are considered SACs because of their ability to interfere and with modifying surfaces. Because these biomolecules are amphiphilic and are produced by different microorganisms, they have different physicochemical properties and physiological roles, which contribute to their specific functions in nature and biotechnological applications [3].

Recently, the production of SACs has received extensive attention because of their diverse applications, such as dissolving water-insoluble compounds, heavy metal binding, contaminant desorption, inhibiting bacterial pathogenesis, adhesion, and cell aggregation [4,5,6,7,8]. In addition, SACs also have several advantages over synthetic surfactants, such as low toxicity, lower critical micelle concentration (CMC), higher biodegradability, and ecological acceptability [9].

Moreover, these compounds exhibit antibacterial [5,10], antifungal [11], antiviral [12], and antitumor activities [13]. Their antiadhesive properties and antibiofilm activities are also important in inhibiting the adhesion and colonization of pathogenic microorganisms and removing preformed biofilms on silicone discs and other biomedical instruments [14].

The present use of these biomolecules has aroused interest from several sectors because of their numerous functions and sustainable properties, allowing various applications in petroleum, food, medicine, pharmaceuticals, chemicals, pulp and paper, textiles, and cosmetics. Furthermore, because of their application in soil bioremediation, they are considered the “green molecules” of the 21st century [15].

## 2. Surface-Active Compounds

### 2.1. Biosurfactants

Biosurfactants (BSs), which are low molecular weight microbial compounds, are synthesized extracellularly or linked to the cell membrane of bacteria [16], yeasts [17], and filamentous fungi [18]. Produced under various extreme environmental conditions, their chemical compositions depend on the microorganism that produces them, raw materials, and process conditions [6].

Surfactants are amphiphilic molecules with a hydrophobic moiety comprising a hydrocarbon chain with saturated or unsaturated and hydroxylated fatty alcohols or fatty acids, and a hydrophilic moiety comprising hydroxyl, phosphate, or carboxyl groups, or carbohydrates (such as mono-, oligo-, or polysaccharides) or peptide fractions [3,19]. Depending on their biochemical nature, these compounds are classified as glycolipids, lipopeptides, lipoproteins or fatty acids, and phospholipid polymers, with glycolipids and lipopeptides being the most abundant [20,21].

Glycolipids consist of mono- or oligosaccharides and lipids, where different sugars (glucose, mannose, galactose, glucuronic acid, or rhamnose) link to form saturated or unsaturated fatty acids, hydroxylated fatty acids, or fatty alcohols. The most studied groups include sophorolipids (SLs), mannosylerythritol lipids, trehalolipids, and rhamnolipids (RLs) [22,23], which are usually produced by the yeast *Starmerella bombicola* [24], *Pseudozyma* sp. [25,26] *Rhodococcus erythropolis* [27] and *Pseudomonas aeruginosa* [28], respectively.

On the other hand, lipopeptides (LP) consist of cyclopeptides with amino acids linked to fatty acids of different chain lengths [29]. The most common among these are surfactin, iturin, and fengycin [29,30,31] which are produced by different microorganisms, such as the genera *Bacillus* [32], in turn, other lipopeptides have been detected in *Bacillus amyloliquefaciens* [33], *Streptomyces* sp. [34], *Pseudomonas*
*guguanensis* [35], and *Serratia marcescens* [36].

Microorganisms that produce BS inhabit water (fresh, underground, and sea) and land (soil, sediments, and mangroves) and can grow in extreme environments (oil reservoirs) and under different temperatures, pH values, and salinity levels [37,38,39].

These microorganisms are generally heterotrophs that need carbon, nitrogen, minerals, vitamins, growth factors, and water to grow and produce metabolites. In general, carbon sources (carbohydrates, oils, and fats) and hydrocarbon groups are often consumed during BS production. For example, glucose, a carbon source easily metabolized by microorganisms through glycolysis to generate energy, is commonly reported as a factor in producing higher yields [37,40].

Because of their amphipathic nature, BSs can mix immiscible fluids, reduce surface and interfacial tensions, and promote solubility of polar compounds in nonpolar solvents [41] that help exhibit numerous properties, such as foaming, dispersion, wetting, emulsification, demulsification, and coating, making them suitable for physicochemical and biological remediation technologies of organic and metallic contaminants [42].

Biosurfactants due to their physicochemical properties have industrial applications in pharmaceuticals, textile processing, agriculture, cosmetics, personal care, and the food industry, as well as environmental applications in soil remediation, hydrocarbon degradation, and oil recovery [43,44,45].

Several BSs have antibacterial, antifungal, antiviral, or antitumor properties, making them potential alternatives to conventional therapeutics in many biomedical applications [45,46].

Despite their versatility and efficiency in terms of applicability in different fields, their production has always been a challenge because of inefficient bioprocessing and high costs due to the expensive substrates used [33]. Therefore, optimizing strategies on cost efficiency and high-yield bioprocessing is critical for low-cost production and mass commercialization.

### 2.2. Bioemulsifier (BE)

Unlike BSs, bioemulsifiers (BEs) have high molecular weight and can emulsify, even at low concentrations, two immiscible liquids, while not reducing surface or interfacial tension [47]. These comprise complex mixtures of heteropolysaccharides, lipopolysaccharides, proteins, glycoproteins, or lipoproteins, which guarantee better emulsification potential and emulsion stabilization [3,48,49].

Bioemulsifiers, which are synthesized by bacteria, yeasts, and filamentous fungi, can be isolated from contaminated soil, mangroves, seawater, freshwater, and human skin [50,51,52,53]. The most studied polymeric BEs include emulsan, alasan, liposan, mannoprotein, and other polysaccharide-protein complexes. Members of the genus *Acinetobacter* sp. are commonly reported to produce BEs [15].

Despite numerous reports on the production of BEs and BSs by different bacteria, the genus *Acinetobacter* spp. received special attention because it is the first known producer of BEs, with emulsan, biodispersan, and alasan as the best examples of BEs commercially produced by the genus. These BEs are mainly used in microbial oil recovery and the biodegradation of toxic compounds [15].

Compared with synthetic surfactants, BEs have many advantages as they are eco-friendly, biocompatible, less toxic with higher biodegradability, and active at extreme temperatures, pH values, and salinity levels. Furthermore, BEs can be produced from low-cost renewable substrates, such as industrial waste, vegetable oils, and hydrocarbons [53].

Various carbon sources are used in BE production, such as ethanol, n-hexadecane, crude oil, glucose, lactic acid, methylnaphthalene, peptone, n-heptadecane, edible oil, olive oil, glycerol, and C-heavy oil [54]. Conventionally, microbial production of BE is still expensive, with the use of synthetic sources as one of the factors contributing most to the high costs. One promising strategy to make the cost economically viable is to include renewable sources from agro-industrial residues and by-products. In this sense, previous research had explored several alternative low-cost substrates, such as residual soybean oil from frying and corn steep liquor, as alternatives to synthetic sources of carbon and nitrogen [53].

Despite their potential advantages, several obstacles hinder practical BE applications, including low yields and high purification costs. To address these issues, researchers have been striving to develop more cost-efficient BEs, which can be used at low concentrations [55].

Bioemulsifiers can form very stable emulsions and dispersions that do not mix, remain attached to the droplet interfaces, and can re-emulsify by adding or replacing a new solvent without diluting. Because of these advantages, BEs are preferred over BSs in the cosmetics and food industries [48].

Because of diverse functions, such as emulsification, wetting, foaming, cleaning, phase separation, surface activity, and hydrocarbon viscosity reduction, BEs are best suited for bioremediation, enhanced oil recovery, cleaning of pipe and vessels contaminated with oil, and more. In addition, emulsifiers are widely used in the food and drug industry [56].

### 2.3. Microorganisms Producing SACS

For many years, researchers have tirelessly searched for microorganisms that have the potential to produce secondary metabolites with surfactant or emulsifying properties. The amount of BS or BE produced depends on the type of microorganisms and their sources (Table 1).

## 3. Biological Properties

### 3.1. Antimicrobial Activities

The discovery of antibiotics in the last century can be considered a major advancement in medicine because the use of these antimicrobial agents significantly reduced morbidity and mortality associated with microbial infections. Antibacterial and antifungal factors reduce and eliminate the viability and growth of microbial populations through several mechanisms: (i) disruption of extracellular membranes and/or their cell wall, (ii) inhibition of gene expression, (iii) DNA damage, or (iv) manipulation of important metabolic pathways [74].

Bacteria become resistant to antimicrobial agents in several ways: through horizontal gene transfer between genetic elements of different strains and the environment that confer resistance and through mutations that interfere with basic cell functions in addition to conferring resistance to microorganisms [75,76].

The most resistant bacteria associated with serious hospital infections include *Enterococcus faecalis*, *Staphylococcus aureus*, *Klebsiella pneumoniae*, *Acinetobacter baumanii*, *P. aeruginosa*, and *Enterobacter* sp., which often result in high mortality rates [77]. Furthermore, other microorganisms such as *Candida* spp. can also be considered a global health threat because of their resistance to antimicrobial agents [78,79,80].

The increasing rates of antimicrobial resistance and the emergence of new microbial pathogens reinforce the need to find new antimicrobial compounds to fight microbial infections. Among these new strategies, SACs have promising antibiotic and disinfectant potential, as well as antibiotic delivery properties due to their physicochemical properties. Most of these biomolecules can break the outer and inner membranes of pathogens, thereby exploiting their charge and hydrophobicity. The advantages of using SACs as antimicrobials include their broad-spectrum bactericidal action and the absence of pathogen resistance mechanisms [81].

Cationic surfactants comprise the largest class of synthetic surfactants with antimicrobial properties because of their broad spectrum of biostatic and biocidal activities against planktonic pathogens. The hydrophobic chain of cationic surfactants penetrates the microbial cell membrane and interferes with membrane continuity and metabolic processes, leading to cell death [82]. Despite exhibiting antimicrobial efficiency mainly against Gram-positive bacteria (29–32 mm), such as *S. aureus* and *Bacillus subtilis*, these compounds are less biodegradable than natural surfactants [83].

Previous studies reported the antimicrobial efficacy of glycolipid SACs produced by microorganisms. For example, RLs produced by *P. aeruginosa* significantly inhibited the growth of *S. mutans* UA159 and *S. sanguinis* ATCC10556. Furthermore, they completely inhibited the growth of *Aggregatibacter actinomycetemcomitans* Y4 at high concentrations [7].

Similarly, the synergistic action of two RL BSs produced by *P. aeruginosa* C2 and *Bacillus stratosphericus* A15 demonstrated bactericidal activity by rupturing the membrane of gram-positive and gram-negative bacteria, such as *S. aureus* ATCC 25923 and *Escherichia coli* K8813 [84]. Because of these actions, the membrane disintegrates, leading to penetration into the cell wall and plasma membrane through the formation of pores, followed by leakage of internal cytoplasmic materials, leading to cell death [85].

A previous study demonstrated that the synergism between essential oils of oregano, cinnamon tree, and lavender with RLs produced by *P. aeruginosa* increased the antimicrobial effect against *Candida albicans* and *S. aureus* which are resistant to methicillin [86], revealing that SAC activity can be enhanced when they establish a synergistic relationship with other compounds. In addition to RLs, SLs are also easily extracted and are usually produced by *Candida* spp. yeast [87] either in the lactone form or the acid form or as a mixture of both forms [88,89].

A previous study showed that SL produced by *C. albicans* SC5314 and *Candida glabrata* CBS138 showed antibacterial properties against pathogenic bacteria *Bacillus subtilis* and *E. coli* [10]. Besides its antibacterial activity against both Gram-positive and Gram-negative bacteria, this class of BS also exhibited promising antifungal activity against pathogenic fungi including *Colletotrichum gloeosporioides*, *Fusarium verticillioides*, *Fusarium oxysporum, Corynespora cassiicola*, and *Trichophyton rubrum* [90].

The antimicrobial activity of SACs glycolipids was found to be dependent on the type of glycolipid and the interaction with the cell membrane. Diaz de Renzo et al. [63] demonstrated that RLs inhibit bacterial growth in the exponential phase while SLs inhibit growth between the exponential and stationary phases.

The antimicrobial potential of lipopeptide SACs has also been recognized; these biomolecules correspond to the most important components of metabolites that are synthesized by many species of the genus *Bacillus* spp., which characterize the strains of this genus as important parts of plant disease control and food safety [91,92,93].

Antimicrobial lipopeptides, such as iturin, fengycin, and surfactin, have been identified in *Bacillus velezensis* HC6. Surfactin exhibited strong antibacterial effects against *Listeria monocytogenes* and *Bacillus cereus*, while fengycin and iturin inhibited the growth of pathogenic fungi *Aspergillus flavus*, *Aspergillus parasiticus*, *Aspergillus sulphureus*, *Fusarium graminearum*, and *Fusarium oxysporum* [94]. Researchers also found that when *B. velezensis* HC6 is applied to corn, it reduced the levels of aflatoxin and ochratoxin produced by fungi.

Ohadi et al. [95] demonstrated that lipopeptides produced by *Acinetobacter junii* displayed nonselective activity against Gram-positive and Gram-negative bacterial strains. The data showed that this bioproduct had effective antibacterial activity at concentrations almost below the CMC and that the minimal inhibitory concentration (MIC) values were lower than the standard antifungal activity, exhibiting almost 100% inhibition against *Candida utilis*.

Other broad classes of bacterial metabolites with surface-active potential and antimicrobial effects include glycoproteins, peptides, and fatty acids. *Lactobacillus* spp. produced a bioactive glycolipoprotein surfactant with antimicrobial activity against *C. albicans* using sugarcane molasses as substrate, and some pathogenic gram-positive and gram-negative bacteria [96]. A cyclic heptapeptide containing a fatty acid moiety produced by *Bacillus subtilis*, called bacaucin 1, exhibited specific antibacterial activity against methicillin-resistant *S. aureus* (MRSA) by disrupting the membrane without detectable toxicity to mammalian cells or induction of bacterial resistance. In addition, this peptide was found to be efficient in preventing infections in both in vitro and in vivo models [97].

Finally, some microorganisms excrete mixtures of bioactive compounds with surface-reducing ability and emulsifying potential. For example, the actinomycete strains of *Streptomyces griseoplanus* NRRL-ISP5009 produced a BS component that is a complex mixture of proteins, carbohydrates, and lipids that have antimicrobial activity against gram-positive bacteria (*Bacillus subtilis*, *S. aureus*) and pathogenic fungi (*C. albicans* and *Aspergillus fumigatus*). However, it is only moderately active against Gram-negative bacteria *E. coli* and *Salmonella typhimurium* [37].

### 3.2. Antiviral Activity

Viruses represent a serious threat to human health at a global level. Previous studies have described secondary metabolites with surface-active properties for their antiviral properties against a variety of viruses. Antiviral activity by SACs was shown to be effective against various viruses, enveloped and nonenveloped (Table 2).

Viral infections represent one of the main causes of human and animal morbidity and mortality that lead to significant healthcare costs [107]. Therefore, secondary metabolites with surface-active properties should be considered promising substances for the development of antiviral compounds.

### 3.3. Anti-Inflammatory Activity

Inflammatory responses represent a crucial aspect of a tissue’s response to certain injuries, chemical irritation, or microbial infections. This complex response involves leukocyte cells, macrophages, neutrophils, and lymphocytes. In response to inflammation, these cells release specialized substances, including amines and vasoactive peptides, eicosanoids, pro-inflammatory cytokines, and acute-phase proteins, which mediate the inflammatory process and prevent additional tissue damage [108].

Currently, studies on SACs are looking into their potential as anti-inflammatory drugs. For example, a recent in vivo study showed that surfactin inhibited the pro-inflammatory response in *Zebrafish larvae* (*Danio rerio*), significantly reducing the expression of interleukin (IL)-1β, IL-8, tumor necrosis factor-α (TNF-α), nitric oxide (NO), nuclear factor kappa-β p65 (NF-κBp65), cyclooxygenase-2 (COX-2), and inducible nitric oxide synthase (iNOS) and increasing the expression of IL-10. Furthermore, the study showed that surfactin reduced neutrophil migration and alleviated liver damage [109].

Other studies showed that surfactin systematically induced CD4 + CD25 + FoxP3 + Tregs in the spleen of mice, which inhibit T cells from producing pro-inflammatory cytokines such as TNF-α and interferon (IFN)-γ. Moreover, surfactin attenuation of chronic inflammation increased IL-10 expression in atherosclerotic lesions of the aorta of mice, demonstrating that BSs can restore the balance in the Th1/Th2 response in mice [110], as well as induce the maturation of dendritic cells (DCs) and increase the expression of MHC-II molecules and other costimulatory factors [111].

Few anti-inflammatory properties related to glycolipid BSs have been reported. Sophorolipids produced by *C. bombicola* reduced lipopolysaccharide-induced expression of TNF-α, COX-2, and IL-6 in RAW 264.7 cells [112], and reduced the level of immunoglobulin E (IgE), TLR-2, IL-6, and STAT3 mRNA expression [113].

In previous in vivo models, SLs reduced sepsis-related mortality and exhibited anti-inflammatory effects in mice by inhibiting nitric oxide and inflammatory cytokine production [114,115]. On the other hand, the glycolipid complex had no significant effect on the proliferative effect of peripheral blood leukocytes because it activated the production of pro-inflammatory cytokines (IL-1β and TNF-α) without affecting the IL-6 production in vitro in the monocyte fraction [116].

### 3.4. Anticancer Activity

Cancer is considered a multistage disease, the etiology of which is associated with high incidence and mortality rates globally. Chemotherapy drugs, surgery, and radiation remain the most common treatments to fight the disease in humans. However, they are all associated with serious adverse effects, indicating the lack of specificity and the need to discover new antitumor agents to improve the effectiveness of conventional chemotherapy drugs while reducing the adverse effects [74].

For these purposes, several studies have demonstrated the antitumor potential of several SACs (Table 3). Biosurfactants have been proposed as suitable drug candidates for many diseases including cancer [117]. Given their wide applications, the interest in exploring their role in promoting human health continues to grow.

### 3.5. Antibiofilm Activity

Biofilms comprise microbial communities attached to the surface and embedded in an extracellular matrix composed of extracellular polymeric substances (EPS) secreted by cells that reside within them. In general, EPS is a mixture of polysaccharides, proteins, extracellular DNA (eDNA), and other smaller components. The biofilm matrix constituents’ physical and chemical properties enable the matrix to protect resident cells from desiccation, chemical disturbance, invasion by other bacteria, and death from predators. However, biofilms often cause major medical problems and are the cause of chronic infections because biofilm communities can house bacteria that are tolerant and persistent against antibiotic treatment and are more resistant to antibiotics compared with planktonic bacteria [9,122].

Because of their composition, biofilms cause a wide range of chronic diseases due to the emergence of antibiotic-resistant bacteria that have become difficult to treat effectively. To date, available antibiotics are ineffective in treating these biofilm-related infections because of their higher MIC and minimal bactericidal concentration values, which may lead to in vivo toxicity. Therefore, designing or tracking antibiofilm molecules that can effectively minimize and eradicate biofilm-related infections is important [123].

Because of their antimicrobial, antiadhesive, and antibiofilm properties, SACs can be used to neutralize biofilm formation and the emergence of drug-resistant microorganisms [14]. These biomolecules tend to interact with antimicrobials [124,125], which are usually less effective against biofilms in general and multispecies biofilms associated with extremely complicated polymicrobial infections.

A mixture of lipopeptides (surfactin, iturin, and fengycin), which are synthesized by *B. subtilis*, prevented biofilm formation by inhibiting cell adhesion of *Trichosporon* spp. by up to 96.89% and dispersed mature biofilms (up to 99.2%), reducing their thickness and cell viability. This mixture reduced cell ergosterol content and altered the membrane permeability and surface hydrophobicity of planktonic cells [126].

Another mixture of lipopeptides (surfactin, iturin, and lichenysin) was identified for the first time in *Lactobacillus* spp. vaginal exhibited strong antiadhesive activity (up to 74.4%) against the biofilm producer *C. albicans* [67]. Mixed lipopeptides (iturin, fengycin, and surfactin) with higher surfactin content produced by *B. subtilis* TIM10 and *B. vallismortis* TIM68 inhibited the biofilm formation of *Malassezia* spp., especially TIM10, by about 90% [127].

Meanwhile, surfactin-type BS produced by *B. subtilis* reduced adhesion and stopped the formation of *S. aureus* biofilm on glass, polystyrene, and stainless-steel surfaces. Surfactin significantly decreased biofilm percentage and reduced *icaA* and *icaD* expressions, which are important for staphylococcal biofilm structure. Furthermore, lipopeptides have been shown to affect the quorum sensing (QS) system in *S. aureus* by regulating the autoinducer 2 activity [94].

In terms of the antibiofilm activity of glycolipids, Allegrone et al. [128] reported the effects of different types of RLs. They demonstrated that RL produced by *P. aeruginosa* 89 (R89BS) was 91.4% pure and comprised 70.6% of monorhamnolipids and 20.8% of dirhamnolipids. The pure extract R89BS inhibited S. aureus adhesion (97%) and biofilm formation (85%). Furthermore, purified monorhamnolipids (mR89BS) have been observed to induce dispersion of preformed biofilms at all concentrations (0.06–2 mg/mL) by 80%–99%, unlike the pure extract R89BS and purified dirhamnolipids (dR89BS), which depended on the tested concentration.

Ceresa et al. [5] demonstrated that R89BS-coated silicone elastomeric disks significantly neutralized *Staphylococcus* spp. biofilm formation in terms of accumulated biomass (up to 60% inhibition in 72 h) and cellular metabolic activity (up to 68% inhibition in 72 h). The results suggested that RL coatings may be an effective antibiofilm treatment procedure and represent a promising strategy for preventing infections associated with implantable medical devices.

Shen et al. [129] demonstrated that besides inhibiting the formation of new biofilms, RLs were superior in eradicating mature biofilms formed by *Helicobacter pylori*, *E. coli*, and *Streptococcus mutans* in a dose-dependent manner, compared with other antibacterial agents even at concentrations below minimum inhibitory concentrations (MICs). They can enhance the effect of antimicrobial agents. Sidrim et al. [130] observed that these molecules significantly increased the activity of meropenem and amoxicillin-clavulanate against mature *Burkholderia pseudomallei* biofilms.

Rhamnolipids produced by *P. aeruginosa* SS14 also inhibited planktonic cells of filamentous fungi of *Trichophyton rubrum* and *Trichophyton mentagrophytes*. The formation and rupture of mature biofilms were dose-dependent, with the highest activity observed at concentrations of 2 × MIC against both pathogens [131].

Like RLs, SLs exhibited an effective inhibitory activity against biofilm formation. Ceresa et al. [132] obtained three different SL products: SLA (acid congeners), SL18 (lactonic congeners), and SLV (mixture of acid and lactone congeners), which all showed an inhibitory effect of 70%, 75%, and 80% for *S. aureus*, *P. aeruginosa*, and *C. albicans*, respectively. Using 0.8% *w/v* SLA on pre-coated medical silicone disks reduced *S. aureus* biofilm formation by 75%. In co-incubation experiments, 0.05% *w/v* SLA significantly inhibited *S. aureus* and *C. albicans* from forming biofilms and adhering to surfaces by 90–95% at concentrations between 0.025 and 0.1% *w/v*.

Antibiofilm activities were also demonstrated for BSs produced by probiotics of the genus Bacillus sp. that were isolated from cervicovaginal samples. This bioproduct, called BioSa3, was highly effective in the formation of biofilms of different pathogenic and multidrug-resistant strains, such as *S. aureus* and *Staphylococcus haemolyticus*. The anti-biofilm effect may be related to the ability of BioSa3 to alter the membrane physiology of the tested pathogens to cause a significant decrease in surface hydrophobicity [133].

Thus, SACs are good candidates for the emergence of new therapies to better control multidrug-resistant microorganisms and inhibit infections associated with biofilms, protecting surfaces from microbial contamination.

### 3.6. Wound Healing

Wound healing is an important but complicated process of tissue repair in humans or animals, comprising a multifaceted process organized by sequential and overlapping phases, including hemostasis, inflammation phase, proliferation phase, and remodeling phase [134,135]. Failure of one of these phases caused by a deregulated immune response or insufficient oxygenation impairs the healing process, leading to ulcerative skin defect (chronic wound) or excessive scar tissue formation (hypertrophic or keloid scarring) [136,137].

Treating wounds of different etiologies constitutes an important part of the total health budget, mostly affected by three important cost drivers: curing time, frequency of dressing change, and complications. Moreover, chronic wound infection, one of the leading causes of nonhealing, contributes significantly to rising healthcare costs. Although the treatment of an uncomplicated surgical incision is relatively inexpensive, the costs can increase significantly when infections occur [138].

Biofilms, commonly found in chronic wounds, contribute to infections, causing slower healing. Infections in chronic wounds are usually caused by multiple species [139], with *P. aeruginosa* and *S. aureus* being the most common. Although most microbial communities usually form on the wound’s outer layer, some biofilms are also embedded in deeper layers, such as *P. aeruginosa* biofilms, which are difficult to diagnose via traditional wound smear culture [140,141]. Moreover, antibiotic resistance of bacteria in biofilms is a crucial problem in the management and treatment of chronic wounds [139].

For these reasons, physicians and the scientific community consider the management and treatment of wounds, as well as biofilm prevention, a top priority. In this context, SACs recently emerged as promising agents that promote wound healing with low irritation and high compatibility with human skin [14]. Furthermore, these bioproducts promote fibroblast and epithelial cell proliferation, faster re-epithelialization, and collagen deposition, leading to a faster healing process [142,143].

Surfactin A from *B. subtilis* promotes wound healing and scar inhibition. During the healing process, it up-regulates the expression of hypoxia-inducible factor-1α (HIF-1α) and vascular endothelial growth factor, accelerates keratinocyte migration via mitogen-activated protein kinase (MAPK), and factor nuclear-κB (NF-κB) signaling pathways and also regulates pro-inflammatory cytokine secretion and macrophage phenotypic exchange. Furthermore, surfactin A inhibits scar tissue formation by influencing α-smooth muscle actin (α-SMA) and transforming growth factor (TGF-β) expression [144]. Therefore, the healing potency of the lipopeptides *B. subtilis* SPB1 is due to their antioxidant activity potential revealed in vitro [143].

A previously unknown lipopeptide 78 (LP78) from *S. epidermidis* inhibited TLR3-mediated skin inflammation and promoted wound healing. The skin lesion activated TLR3/NF-κB, promoting p65 and PPARγ interaction in the nuclei and initiating the inflammatory response in keratinocytes. Next, LP78 activated the TLR2-SRC, inducing β-catenin phosphorylation in Tyr. Phospho-β-catenin is translocated into the nuclei to bind to PPARγ, thereby interrupting the p65 and PPARγ interaction. Dissociation between p65 and PPARγ reduced TLR3-induced inflammatory cytokine expression in skin wounds of normal and diabetic mice, which correlated with faster wound healing [145].

As an alternative to improve this healing process, the formulation of nanolipopeptide biosurfactant (NLPB) from the lipopeptide biosurfactant (LPB) produced by *Acinetobacter junii* was reported as promising for performing healing activity. The percentage of wound closure of mice treated with NLPB hydrogels at 2 mg/mL was approximately 80% on day 7 and 100% on day 15. The NLPB hydrogel formulation showed better efficacy in wound closure and healing when compared to the control [146].

A BS of glycolipid nature, which was synthesized by *Bacillus licheniformis* SV1, showed good cytocompatibility and increased 3T3/NIH fibroblasts proliferation in vitro. A previous study showed that the application of BS ointment in a skin excision wound in rats promoted re-epithelialization, fibroblast cell proliferation, and faster collagen deposition, demonstrating its potential transdermal properties to improve skin wound healing [147].

A previous study administered an RL-containing ointment (5 g/L) on the back of Wistar mice after creating an excision wound. Histopathological results revealed a significant healing effect of RL, demonstrating increased wound closure, improved collagenases, and reduced inflammation (decreasing the level of TNF-α) without inducing skin irritation [84]. Dirhamnolipid treatment has been suggested for cutaneous scar therapy, demonstrating an antifibrotic function in rabbit ear hypertrophic scar models with a significant reduction in the scar elevation index, type I collagen fibers, and α-SMA expression [148].

A cell culture model has demonstrated the wound healing capacity of SLs by using an in vitro human dermal fibroblast model as a substitute for human skin, revealing that SLs affected the ability of human skin fibroblasts to express collagen I mRNA (Col-I) and elastase inhibition (IC_50_ = 38.5 μg/mL) [112]. In addition, Kwak et al. (2021), using an in vitro wound healing assay in human colorectal adenocarcinoma (HT-29) cell line, showed a significantly increased collagenase-1 expression (*p* < 0.05) 48 h after SL treatment. Moreover, all SL dosages significantly increased occludin and matrilysin-1 (MMP-7) expression [149].

### 3.7. Other Considerations

We also consider that there are SACs molecules obtained by chemical synthesis processes, such as ultrashort synthetic surface active (USSA) [150,151]. Some of these can be synthesized as C-terminal amides on Rink amide (4-Methylbenzhydrylamine (MBHA) resin using 9-fluorenylmethoxycarbonyl/t-butylcarbamate [151]. The fundamental difference of the USSA, as lipopeptoids (modified SAC) in relation to the natural ones, is their immunomodulatory capacity. As seen in mouse infection models, they reduce the exacerbation of the disease, even if not presenting direct antibacterial activity [151]. This characteristic would be a limiting activity, since many natural ones lead to a disturbance of biological membranes, with antifungal and antibacterial actions [151].

New possibilities can be obtained for the SACs, as transformation systems applying recombinant plasmids have been employed to substantially increase the productivity of microbial biosurfactants, e.g., the engineered strain *Pseudozyma* sp. SY16, which increases the production of mannosylerythritol lipids (MELs) by up to 31.5%, suggesting that genetic engineering can improve the industrial application of yeast [152].

## 4. Conclusions

The BS and BE surface-active compounds have drawn the attention of the scientific community as a new generation of products with high potential in the biomedical and pharmaceutical fields. Their use, whether alone or in combination with other antimicrobial or chemotherapeutic agents, opens paths for new strategies to prevent and combat infections caused by bacteria, fungi, and viruses, as well as the formation and proliferation of biofilms. Furthermore, new anticancer treatments and wound healing applications can be explored in future studies.

These molecules affect various biological activities, making them suitable candidates for use in new generations of agents in the biotechnological, biomedical, and pharmaceutical fields. However, it is necessary to investigate their applications, cost-effectiveness, and availability further.

## Figures and Tables

**Table 1 antibiotics-11-01106-t001:** Lists some microorganisms that produce surface-active compounds.

Microorganism	Biosurfactant/Bioemulsifier	Reference
*Acinetobacter calcoaceticus* RAG-1	Emulsan	[57]
*Acinetobacter radioresistant* KA53	Alasan	[58]
*Acinetobacter junii* B6	Surfactin/fengycin	[59]
*Acinetobacter junii* BD	Rhamnolipids	[60]
*Acinetobacter calcoaceticus* A2	Biodispersan	[61]
*Bacillus nealsonii* strain S2MT	Surfactin	[2]
*Bacillus subtilis* 3NA	Surfactin	[62]
*Bacillus thailandensis* E264	Rhamnolipids	[63]
*Bacillus velezensis*	Iturin, surfactin, and fengycin	[64]
*Candida keroseneae* GBME-IAUF-2	Sophorolipids	[65]
*Candida lipolytica* UCP 0988	Rufisan	[66]
*Lactobacillus* sp.	Surfactin, iturin, and lichenysin	[67]
*Pseudomonas aeruginosa* SG	Rhamnolipids	[68]
*Pseudomonas fluorescens* SBW25	Viscosine	[69]
*Pseudomonas* sp. S2WE	Rhamnolipids	[70]
*Serratia* sp. ZS6 strain	Serrawettina	[71]
*Yarrowia lipolytica* IMUFRJ50682	Yansan	[72]
*Trichosporon mycotoxinivorans* CLA2	Lipid-polysaccharide complex	[73]

**Table 2 antibiotics-11-01106-t002:** Antiviral properties of SACS.

Biosurfactant/ Bioemulsifier	Microorganism	Antiviral Activity	Virus	Reference
Surfactin	*Bacillus subtilis*	Rupturing the viral lipid membrane and part of the capsid	Semliki Forest virus	[98]
			Simplex virus	
			(HSV-1, HSV-2)	
			Suid herpesvirus (SHV-1)	
		Inhibited the proliferation	Simian immunodeficiency (SIV)	[99]
			Feline calicivirus (FCV)	
			Coronaviruses:	
			Epidemic porcine diarrhea (PEDV)	
			Transmissible gastroenteritis virus (TGEV)	
Lipopeptides	-	Inhibited the membrane fusion between the virus and host cells.	Influenza A (H1N1)	[100]
			Human Coronavirus SARS-CoV-2	[101,102,103]
Sophorolipids	*Candida bombicola*	Virucidal property	Human Immunodeficiency Virus (HIV)	[104,105]
Rhamnolipids	*Pseudomonas* spp.	Inhibits the cytopathic effect	Simplex virus:	[106]
			HSV-1, and HSV-2;	
	*Pseudomonas gessardii* M15	Inhibited the proliferation	Simplex vírus:	[12]
			HSV-1 and HSV-2,	
			Human coronavírus:	
			HCoV-229E and SARS-CoV-2	

**Table 3 antibiotics-11-01106-t003:** Anticancer activity of SACS against cancer cells.

Biosurfactant/ Bioemulsifier	Microorganism	Anticancer Activity	Cancer	Reference
Rhamnolipids: monorhamnolipid and dirhamnolipid	*P. aeruginosa* MR01	Inhibiting cell division at lower concentrations	Human breast cancer MCF-7	[118]
Sophorolipids	*Wickerhamiella domercqiae* Y2A	Increased the apoptosis	HepG2 liver cancer cells	[109]
		Cytotoxicity	Breast cancer MDA-MB-231	[119]
		Inhibited cell proliferation	Liver Lung Leukemia	[120]
Surfactin		Reduced tumor growth and weight; Apoptosis; Elevated levels of immune-boosting mediators	Melanoma skin cancer	[13]
	*Bacillus saphensis*	Cytotoxic activity against cancer cell lines	Breast cancer Melanoma	[46]
Iturin	*Bacillus megaterium*	Inhibited the growth of cancer cells	Breast cancer	[121]

## Data Availability

Not applicable.

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
