# Peer review of "Surface-Active Compounds Produced by Microorganisms: Promising Molecules for the Development of Antimicrobial, Anti-Inflammatory, and Healing Agents"

_antibiotics, 2022, doi:10.3390/antibiotics11081106_

Round 1
Reviewer 1 Report
Really nice review, well written and well explained!
Author Response
No reviewer suggestions.
Reviewer 2 Report
The manuscript is a review on the biotechnological and biomedical properties of Surface-active compounds (SACs), an extensive bibliographic review is observed, the manuscript is well organized with good writing and good English. All scientific names (either genus and/or species) must be in italics, the entire manuscript must be reviewed.
Author Response
Point
1) All scientific names (either genus and/or species) must be in italics, the entire manuscript must be reviewed.
R-The requested changes have been included in the manuscript
Reviewer 3 Report
I have found this article very interesting, comprehensive and scientifically sound.
Author Response
No reviewer suggestions
Reviewer 4 Report
The review from Araujo et al., focused on surface active compounds produced by microorganisms and their applications. The present article is well written and detailed covering applications mainly from pharmaceutical/medicine perspective. The review provides the readers in-depth with the updated information on surface active compounds in one place rather than multiple articles. Overall, this review is fit for publication. However, there are some major and minor issues in the current manuscript.
Major Comments:
- The content of manuscript does not justify the title of the manuscript completely. Biological applications is a broader term that covers use of microbes in wide range of fields such as food industry, bioremediation etc. The manuscript mostly focuses on the use of surface active compounds in reference to its applications in the field of therapeutics covering antibacterial, antiviral, anti-inflammatory etc. The title can be misleading for reader and either the content or title should be modified accordingly.
- Authors also need to discuss short or ultrashort synthetic surface active compounds and its key design feature for comparison with biologically derived compounds to provide an insight whar are the key limitations and challenges in both approaches.
- The authors also need to discuss importance of the Genetically engineered microorganisms in the bio-based synthesis of surface active compounds since most of microorganisms does not produce surface active compounds on their own.
- The current manuscript suffers from its lengthiness and there is too much information to read. I will recommend the authors to include information in form of tables and figures It would be more convenient for the reader to compare the BS/BE functions (microbes) looking at the tables and figures.
- The content appears repetitive at many places. For example, LineLine 17-19 and 32-34 proved same information. Also, Line 19-20 and Line 46-48 (line 101-103); Line 43 – 45 and Line 94-96. Please check throughout to avoid repetition.
General comments:
- The bacterial name should be Italic. For example, Line 239, Line 253. Check throughout the manuscript.
- Include list of abbreviations similar to keywords.
- There are some typographical errors that needs to be checked.
Author Response
Response and comments
Point 1. The content of manuscript does not justify the title of the manuscript completely. Biological applications is a broader term that covers use of microbes in wide range of fields such as food industry, bioremediation etc. The manuscript mostly focuses on the use of surface active compounds in reference to its applications in the field of therapeutics covering antibacterial, antiviral, anti-inflammatory etc. The title can be misleading for reader and either the content or title should be modified accordingly.
Response 1: The suggestion was accepted and the title of the manuscript changed to “Surface-active compounds produced by microorganisms: promising molecules for the development of antimicrobial, anti-inflammatory, and healing agents”
Point 2. Authors also need to discuss short or ultrashort synthetic surface active compounds and its key design feature for comparison with biologically derived compounds to provide an insight whar are the key limitations and challenges in both approaches.
Response 2: We agree with the suggestion and add in other considerations
Point 3. The authors also need to discuss importance of the Genetically engineered microorganisms in the bio-based synthesis of surface active compounds since most of microorganisms does not produce surface active compounds on their own.
Response 3: We agree with the suggestion and add in other considerations
Point 4. The current manuscript suffers from its lengthiness and there is too much information to read. I will recommend the authors to include information in form of tables and figures It would be more convenient for the reader to compare the BS/BE functions (microbes) looking at the tables and figures.
Response 4: We agreed with the suggestion and added tables in the sections Anticancer activity and Antiviral activity
Point 5. The content appears repetitive at many places. For example, LineLine 17-19 and 32-34 proved same information. Also, Line 19-20 and Line 46-48 (line 101-103); Line 43 – 45 and Line 94-96. Please check throughout to avoid repetition.
Response 5: we agree with the items and modify the text in the manuscript

Round 2
Reviewer 4 Report
The authors have made amendments in response to my previous review and provided response to each question raised by me to my satisfaction. Therefore, I endorse the manuscript for publication in its current form.